# Optimization of the Hydrodynamic Performance of a Double-Vane Otter Board Based on Orthogonal Experiments

**Lei Wang** [1,2,*], **Xun Zhang** [2], **Rong Wan** [1,*], **Qingchang Xu** [3] **and Guangrui Qi** [2]

1. College of Marine Sciences, Shanghai Ocean University, Shanghai 201306, China
2. Key Laboratory of Oceanic and Polar Fisheries, Ministry of Agriculture, East China Sea Fisheries Research Institute, Chinese Academy of Fishery Sciences, Shanghai 200090, China
3. Key Laboratory of Sustainable Development of Marine Fisheries, Ministry of Agriculture and Rural Affairs, Yellow Sea Fisheries Research Institute, Chinese Academy of Fishery Sciences, Qingdao 266071, China
* Correspondence: emperor0228l@163.com (L.W.); rongwan@shou.edu.cn (R.W.); Tel.: +86-137-8899-5939 (L.W.)

**Abstract:** An orthogonal experiment was conducted to study the effects of the aspect ratio, camber of the fore wing, and gap ratio of front and rear wing panels on the hydrodynamic performance of double-vane otter boards. The design of the orthogonal experiment comprised three factors (each having three levels), namely the aspect ratio (1.0, 1.5, and 2.0), camber (0.12, 0.14, and 0.16), and gap ratio (0.25, 0.30, and 0.35), the drag coefficient $C_x$ and the lift coefficient $C_y$ of nine otter board models obtained in a wind tunnel experiment, and the lift–drag ratio $K$ obtained by calculation. The lift–drag ratio for a working angle of attack of 30° was selected as the inspection index, and the experimental data were analyzed in an orthogonal design-direct analysis. Analysis of each factor revealed that the optimal level combination of factors was $A_3B_3C_3$ and that the decreasing order of the effects of the factors was A (aspect ratio) > B (gap ratio) > C (camber). The orthogonal experiment thus obtained an optimal otter board in terms of the aspect ratio (2.0), fore wing camber (0.16), and gap ratio (0.35), with the aspect ratio having the greatest effect on performance. The hydrodynamic performances of the otter board with the optimized structure and another otter board model were compared in numerical simulation, which verified the correctness of the analysis results. The experimental results provide a reference for the optimal design of the double-vane otter board.

**Keywords:** otter board; hydrodynamic; wind tunnel; orthogonal experiment; trawl

## 1. Introduction

An otter board is a piece of rising equipment that is assembled on both sides of a trawl mouth and uses the horizontal expansion force generated when moving forward through the water to expand the net mouth. There are diverse types and structures of otter boards, and the purpose of performance optimization is to increase the expansion force of the otter board and reduce the resistance of the otter board in improving the fishing production efficiency of the trawl. The design of the double-vane otter board is based on the concept of the biplane, and the double-vane otter board is thus also called the biplane trawl door. Studies on biplanes show that they have a better overall aerodynamic efficiency and lift–drag ratio compared with monoplanes [1,2]. Similar to the structure of a biplane, the double-vane otter board has front and rear panels. The aspect ratio, gap, and staggering angle of the wings are important factors affecting the performance of biplanes. Additionally, the camber of the panel is an important factor affecting the performance of the otter board. Scholars have carried out a series of studies on the double-vane otter board. Fukuda et al. [3–5] compared the hydrodynamic performances of double-vane otter boards and single-wing otter boards and investigated the effects of the wing gap and staggering angle on the hydrodynamic performance of double-vane otter boards. Takahashi et al. [6] adopted computational fluid dynamics and tank test methods in studying the effect of the gap in double-vane otter boards. The two methods provide basically the same results for

the effect on the hydrodynamic performance of the otter board. Mellibovsky et al. [7,8] investigated the hydrodynamic performance of a mesh plate in a flume experiment and wind tunnel experiment and carried out a verification analysis in a sea experiment. Although the hydrodynamic changes in the wind tunnel experiment and flume experiment were consistent, they recommended the wind tunnel experiment as the better research method because of its higher efficiency and accuracy. Zhuang et al. [9] investigated the performance optimization of the biplane otter board by designing the wing spacing, aspect ratio, and staggering angle of the otter board and found that the biplane otter board had a better lift and a greater stall angle than the monoplane otter board. You et al. [10] proposed a new type of double-wing trawl gate with an aspect ratio of 2.0 and a camber of 20%. They found that the selectable working angle of attack with a lift coefficient greater than 2.0 has a wider range and better stability.

The above-mentioned research reveals that the double-vane otter board is an important concern for the performance optimization of the otter board. Various factors, such as the wing shape, wing camber, wing gap, staggering angle of the wings, and aspect ratio, affect the performance of the double-vane otter board. The most researched method of constructing the double-vane otter board is to use two panels of the same structure and adjust the relative positions (e.g., the spacing and staggering angle) and shapes of the panels. Adopting this method, the high-lift biplane otter board has been optimized. Referring to the parallel-wing structure of aircraft, the present study adopted parallel wings for the design of a double-vane otter board (where two-layer curved plates are concentric circles) in exploring a new method of designing the double-vane otter board. By adjusting the aspect ratio, camber, and gap ratio, the present study conducted an orthogonal experiment on the double-vane otter board.

A wind tunnel test is an important method of studying the hydrodynamic performance of an otter board in that it can quickly and accurately obtain various hydrodynamic performance parameters. Additionally, with the application and development of computational fluid dynamics software, numerical simulation has become an important method of studying the hydrodynamic performance of the otter board. Comparing the results of numerical optimization operations with model experimental data can greatly save time and costs. Referring to the above methods of researching the hydrodynamic performance of the otter board, the present study obtained the hydrodynamic performance parameters of nine models in a wind tunnel experiment. The effects of factors on the hydrodynamic performance of the otter boards were then systematically and comprehensively analyzed and the optimization results were verified in numerical simulation. The otter board structure was thus optimized, providing a reference for research on the structural design and performance of otter boards.

## 2. Materials and Methods

### 2.1. Design and Manufacture of the Otter Board Model

The basic structure of the test model is a double-vane curved otter board, which is mainly suitable for mid-water trawl operations. The frontal projected area of an otter board used in mid-water trawl operations is generally 4–8 m$^2$. The structure and parameters of the model used in the present study are illustrated in Figure 1.

The otter board comprises two layers of fore and rear wing panels. The arcs of the two layers of wing panels are designed as parts of concentric circles, which refers to the parallel wing arrangement of biplanes [1]. The chord length and curvature of the fore wing panel are designed to determine the size of the fore wing panel. The projection length of the slit, the front staggering angle, and the tail staggering angle are adjusted to obtain the desired size of the rear wing panel.

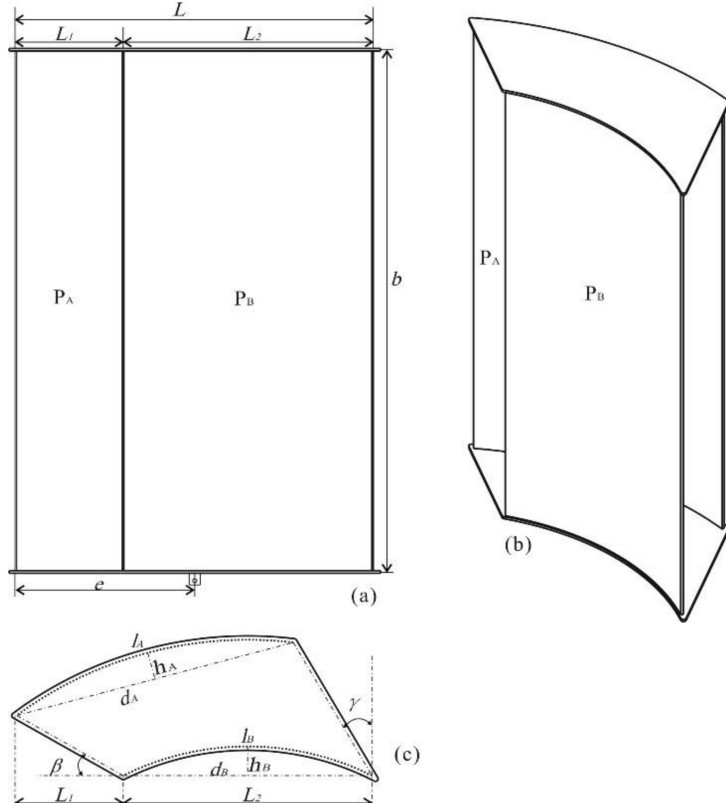

**Figure 1.** Structure and parameters of the otter board model. (**a**) Front view; (**b**) Configuration; (**c**) Top view. $P_A$: rear wing; $P_B$: fore wing; $L$: chord; $b$: span; $l_A$, $l_B$: arc lengths of panels; $L_1$: projection length of the gap; $L_2$: projection length of $l_B$; $h_A$, $h_B$: distances from the midpoint of the arc to the chord; $d_A$, $d_B$: chord lengths of panels; $e$: distance from the leading edge to the center pivot; $\beta$: front staggering angle (i.e., the angle formed by the fore wing chord and the line connecting with the left edge of both wings); $\gamma$: tail staggering angle (i.e., the angle formed by the line connecting with the right edge of both wings and a vertical line).

According to the structural design of an otter board, the present study considered three factors of the hydrodynamic performance of the otter board, namely the gap ratio of the otter board ($L_1/L$), the camber of the fore wing panel ($h_B/d_B$) and the aspect ratio ($b/L$). Each factor was set at three levels to determine its effect on performance. The selection of values refers to existing research and the application requirements of an otter board in mid-water trawling. The gap ratio was set at 0.25, 0.30, and 0.35, the wing panel camber was set at 0.12, 0.14, and 0.16, and the aspect ratio was set at 1.0, 1.5, and 2.0. An $L_9$ ($3^4$) orthogonal experiment was designed as presented in Table 1 and models were constructed as presented in Table 2 to determine the best combination of factors [11].

**Table 1.** Factors of the orthogonal experiment.

| Level | Factor | | |
|---|---|---|---|
| | **Aspect Ratio (A)** | **Camber (B)** | **Gap Ratio (C)** |
| I | 1.0 | 0.12 | 0.25 |
| II | 1.5 | 0.14 | 0.30 |
| III | 2.0 | 0.16 | 0.35 |

**Table 2.** Factor combinations of the otter board model.

| No. | Level | Aspect Ratio | Level | Camber | Level | Gap Ratio |
|-----|-------|--------------|-------|--------|-------|-----------|
| 1 | I | 1.0 | I | 0.12 | I | 0.25 |
| 2 | I | 1.0 | II | 0.14 | II | 0.30 |
| 3 | I | 1.0 | III | 0.16 | III | 0.35 |
| 4 | II | 1.5 | I | 0.12 | II | 0.30 |
| 5 | II | 1.5 | II | 0.14 | III | 0.35 |
| 6 | II | 1.5 | III | 0.16 | I | 0.25 |
| 7 | III | 2.0 | I | 0.12 | III | 0.35 |
| 8 | III | 2.0 | II | 0.14 | I | 0.25 |
| 9 | III | 2.0 | III | 0.16 | II | 0.30 |

The main dimensions of the otter board model ($L$ and $b$) need to be designed with reference to the wind tunnel experimental conditions and model test similarity criteria. According to the experimental conditions of the wind tunnel, the effect of the wall surface layer of the wind tunnel shall be considered when the otter board model is in the wind tunnel test section, and the degree of obstruction of the model shall not exceed 5%, and the height shall not exceed 30% of the height of the test section [12]. The test section of the wind tunnel in the present experiment had a height of 2.5 m and a cross-sectional area of 7.18 m$^2$. It follows that $b \leq 0.75$ m and $L \cdot b \leq 0.035$ m$^2$, and the chord length $L$ of the nine otter board models is thus designed as $L = 500$ mm ($b/L = 1.00$), 400 mm ($b/L = 1.50$) and 300 mm ($b/L = 2.00$).

The front and back staggering angles of the double-layer wing were set as $\beta = \gamma = 30°$. The structures of the nine otter board models were designed according to the orthogonal factors of the model experiment. The specific structural parameters of each otter board are given in Table 3. The otter board model was constructed from steel, and the surface was painted according to the actual production of a physical otter board, as displayed in Figure 2.

**Table 3.** Dimensions and structural parameters of the otter board models.

| No. | $L$/m | $b$/m | $A$ | $S$/m$^2$ |
|-----|-------|-------|-----|-----------|
| 1 | 0.50 | 0.50 | 1.0 | 0.25 |
| 2 | 0.50 | 0.50 | 1.0 | 0.25 |
| 3 | 0.50 | 0.50 | 1.0 | 0.25 |
| 4 | 0.40 | 0.60 | 1.5 | 0.24 |
| 5 | 0.40 | 0.60 | 1.5 | 0.24 |
| 6 | 0.40 | 0.60 | 1.5 | 0.24 |
| 7 | 0.30 | 0.60 | 2.0 | 0.18 |
| 8 | 0.30 | 0.60 | 2.0 | 0.18 |
| 9 | 0.30 | 0.60 | 2.0 | 0.18 |

Note: $L$: chord; $b$: span; $A$: $b^2/S$, aspect ratio; $S$: $L \times b$, the projected area of the otter board model.



**Figure 2.** Nine otter board models: 1–9, corresponding to the numbers of nine otter board models in Table 3.

### 2.2. Test Facility

The experiment was conducted in the NH–2 wind tunnel at Nanjing University of Aeronautics and Astronautics, China. Dimensions of the wind tunnel test section were 6 m (length) × 3 m (width) × 2.5 m (height). The cross-section of the wind tunnel, as shown by the outer contour in Figure 3a, had an area of 7.18 m². Figure 3 illustrates the experimental setup inside the wind tunnel. The model otter board was attached to a six-component strain-gage balance to measure forces in all directions.

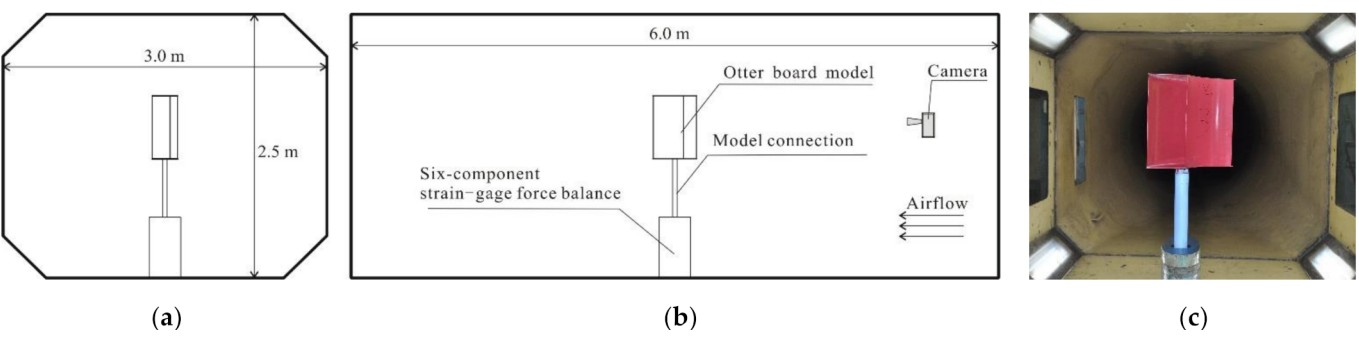

**Figure 3.** Otter board model in the wind tunnel: (**a**) Left side view; (**b**) Front view; (**c**) Photograph taken on the right in the wind tunnel before testing; there was no camera when testing.

### 2.3. Test Method

2.3.1. Parameter Definitions of the Test Model

The test model was installed on the mechanical base of the six-component strain-gage balance in the wind tunnel. Measurements were conducted as follows. The angle of attack of the model was rotated from 0°–70° when the wind speed reached 28 m·s⁻¹ (at room temperature of 20 °C). Measurements were made at 2.5° intervals when the angle of attack was in the range of 0°–60° and at 5° intervals when the angle of attack was greater than 60°.

Parameters of the models in the wind tunnel test section are defined in Figure 4. In the figure, $O$ is the torque reference point, where the bottom of the model connects to the model connection. In the test, the resistance of the model was provided by the balancing force in the $X$–direction and the lift was provided by the balancing force in the $Y$-direction.

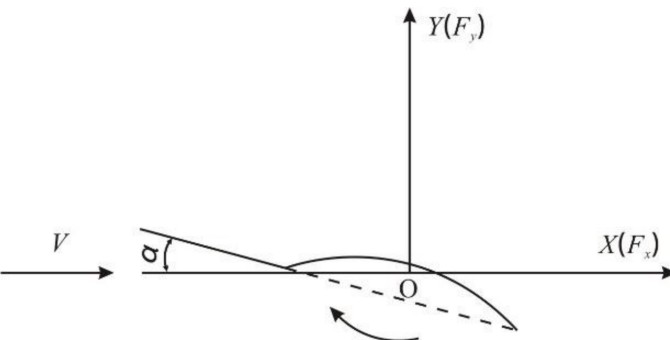

**Figure 4.** Parameter definitions of the test model in the wind tunnel. $F_y$: lift; $F_x$: drag; $V$: wind speed; $\alpha$: angle of attack.

The Reynolds number of the otter board is calculated [13] as

$$R_e = VL/\nu. \tag{1}$$

In the present experiment, the wind speed $V = 28\ \mathrm{m\cdot s^{-1}}$ and the coefficient of viscosity $\nu = 15 \times 10^{-6}\ \mathrm{m^2\cdot s^{-1}}$. Equation (1) gives a Reynolds number $R_e$ of the nine otter boards under the above conditions between $0.57 \times 10^6$ and $0.94 \times 10^6$, which indicates that this experimental design is in the automatic model area of the model test [14].

### 2.3.2. Data Processing

The drag coefficient $C_x$, lift coefficient $C_y$ and lift–drag ratio $K$ are expressed [15] as

$$C_x = 2\cdot\rho^{-1}\cdot S^{-1}\cdot V^{-2}\cdot F_x, \tag{2}$$

$$C_y = 2\cdot\rho^{-1}\cdot S^{-1}\cdot V^{-2}\cdot F_y, \tag{3}$$

$$K = C_y/C_x, \tag{4}$$

where $F_y$ and $F_x$ are the measured lift and drag forces (N), respectively, $\rho$ (kg·m$^{-3}$) is the air density, $S$ is the projected area of the otter board (m$^2$), and $V$ is the actual velocity of incoming wind (m·s$^{-1}$).

### 2.3.3. Modeling and Meshing in Numerical Simulation

In the calculations, the fluid flow was assumed to be that of an isotropic incompressible fluid, and the steady Reynolds-averaged Navier–Strokes (RANS) equations were taken to be the governing equations for describing the fluid flow [16]:

$$\frac{\partial \overline{u}_i}{\partial x_i} = 0, \tag{5}$$

$$\frac{\partial \overline{u}_i}{\partial t} + \overline{u}_j\frac{\partial \overline{u}_i}{\partial x_j} = -\frac{1}{\rho}\frac{\partial \overline{p}}{\partial x_i} + \frac{\partial}{\partial x_j}\left(\nu\frac{\partial \overline{u}_i}{\partial x_j}\right) - \frac{\partial \overline{u_i u_j}}{\partial x_j}, \tag{6}$$

where $\overline{u}_i$ denotes the time-averaged velocities of the RANS model, $u_i u_j$ denotes the Reynolds stresses denoting fluctuating components (with $i$ and $j$ representing different flow directions), $t$ is time (s), $\overline{p}$ is the pressure (N), $\rho$ is the density (kg·m$^{-3}$), and $\nu$ is the kinematic viscosity.

The $k$–$\varepsilon$ explicit algebraic Reynolds stress model (EARSM) was adopted as the turbulence model in the simulation. Combining the standard $k$–$\varepsilon$ turbulence model with an EARSM can improve the accuracy of hydrodynamic loading, as demonstrated by Wang [17,18]. A scalable wall treatment was adopted for the wall function [19].

The optimal results of the orthogonal experiment were analyzed and verified in numerical simulation adopting the numerical computational fluid dynamics code ANSYS–

CFX 19.0. The simulation conditions were consistent with the experimental conditions. The test section was taken as the 6.0 m × 2.5 m × 3.0 m calculation domain in the wind tunnel simulation (Figure 5). The inlet and outlet were defined as the flow inlet and outlet of the computational domain in the x-direction, respectively. $k$ and $\varepsilon$ were initialized as turbulence intensities following Wang et al. [20]. The fluid medium used in the numerical simulation was incompressible air. Assuming that the fluid could be completely discharged, the outlet boundary was the pressure outlet, and the relative pressure was 0 Pa. Furthermore, all surfaces of the otter board and other surfaces of the domain were assumed to be no-slip walls. The current velocity at the inlet boundary was set at a uniform 28.0 m·s$^{-1}$ in the x-direction, the turbulence intensity was 2.86–3.05%, and the intensity and viscosity ratio was 772–1331. Table 4 gives the parameter settings used in the numerical simulation.

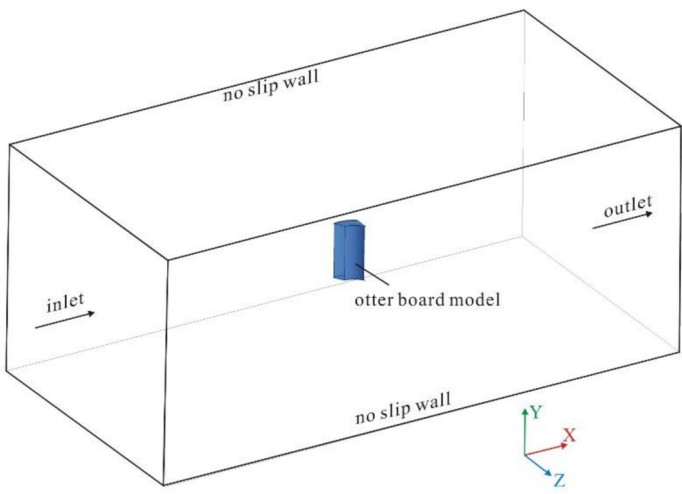

**Figure 5.** Boundary conditions of the computational domain.

**Table 4.** Parameter settings of the numerical simulation.

| Setting Items | Options |
| --- | --- |
| Simulation type | 3D, Steady |
| Solver | CFX, Double precision, Pressure-based and implicit |
| Turbulence model | k–$\varepsilon$ EARSM model |
| Pressure | Standard |
| Inlet | Velocity inlet |
| Outlet | Pressure outlet |
| Otter board | No-slip wall |
| Wall boundary condition | Scalable wall function |

In the calculations, the grid was generated as an unstructured mesh, and the mesh around the otter board was refined locally. To improve the accuracy of numerical simulations, a sensitivity analysis of the mesh resolution was performed (sim-1 to sim-5, Table 5). The drag coefficient and lift coefficient were selected as indicators in the analysis. The two indicators stabilized (varying by less than 1%) when the number of elements reached $4.5 \times 10^6$. Referring to Wang et al. [18], $4.5 \times 10^6$ elements are sufficient to ensure the accuracy of the numerical simulation. In subsequent numerical simulations of the present study, the number of cells was set at approximately 4.5 million. The solver Yplus around the otter board was in the range of 11.06–89.01. Figure 6 shows a cross-sectional view of the meshing around the otter board.

**Table 5.** Sensitivity analysis of the mesh resolution.

| No. | $C_x$ | $C_y$ | K | Nodes | Elements |
|---|---|---|---|---|---|
| Sim-1 | 0.532 | 1.718 | 3.232 | $2.4 \times 10^5$ | $1.3 \times 10^6$ |
| Sim-2 | 0.542 | 1.735 | 3.204 | $3.1 \times 10^5$ | $1.7 \times 10^6$ |
| Sim-3 | 0.559 | 1.742 | 3.115 | $4.4 \times 10^5$ | $2.4 \times 10^6$ |
| Sim-4 | 0.566 | 1.766 | 3.122 | $8.5 \times 10^5$ | $4.5 \times 10^6$ |
| Sim-5 | 0.565 | 1.760 | 3.115 | $1.4 \times 10^6$ | $7.6 \times 10^6$ |
| * Exp | 0.600 | 1.843 | 3.072 | | |

* Experiment data.

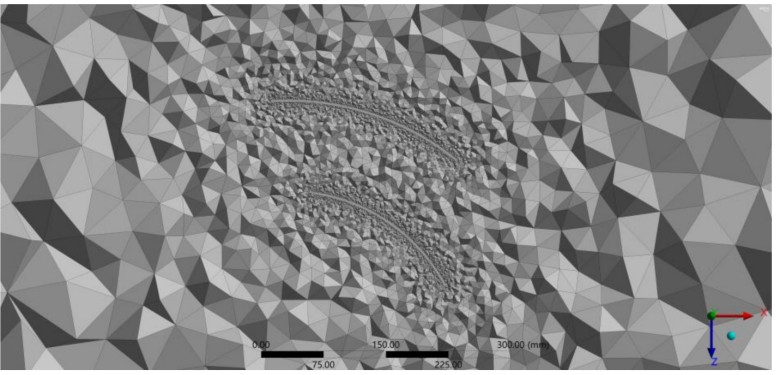

**Figure 6.** The meshing of an otter board in numerical simulation.

The simulation was conducted at 2.5° intervals of the angle of attack of the otter board in the range of 0°–60°. The outputs of the simulation were the drag and lift of the model. The drag coefficient $C_x$, lift coefficient $C_y$ and lift–drag ratio $K$ were calculated.

## 3. Results

### 3.1. Drag Coefficient, Lift Coefficient and Lift–Drag Ratio of the Otter Board

Figure 7 presents curves of the drag coefficient $C_x$ versus the angle of attack $\alpha$ of the incoming flow for the nine otter board models. The relationship between $C_x$ and $\alpha$ is largely linear but changes at an angle of attack of 40°–55°. In this angle range, there is a resistance-falling inflection zone except for otter board models 3 and 7.

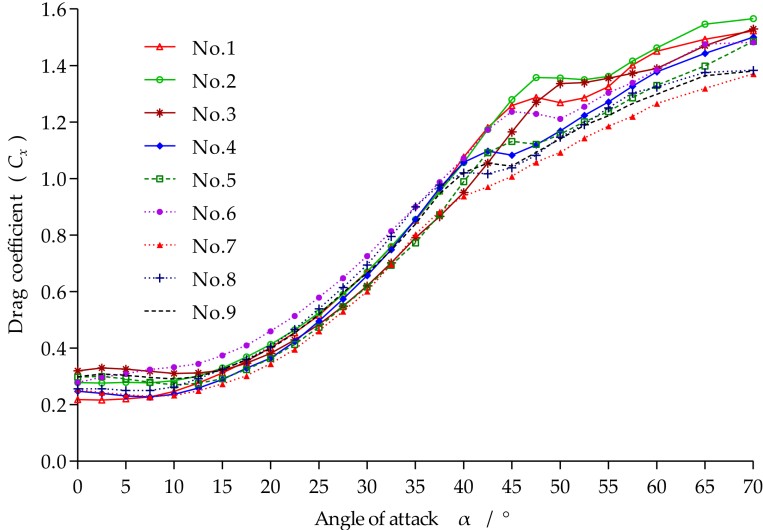

**Figure 7.** $C_x$ versus the angle of attack $\alpha$ for the nine otter board models.

Figure 8 presents curves of the lift coefficient $C_y$ versus the angle of attack $\alpha$ of the incoming flow for the nine otter board models. The maximum lift coefficient $C_y$ is highest

for otter board model 6, being 2.221 ($\alpha$ = 37.5°). The maximum lift coefficient is lowest for otter board model 1, being 1.881 ($\alpha$ = 40°). The angle of attack corresponding to the maximum lift coefficient is called the critical angle of attack. The critical angle of attack ranges from 32.5° to 40° across the nine models is the largest for otter board model 3 and is the smallest for otter board models 7 and 8. The curves of the nine otter board models show the same trend of a rapid decline of the lift coefficient within an angle range of approximately 10° after the critical angle of attack.

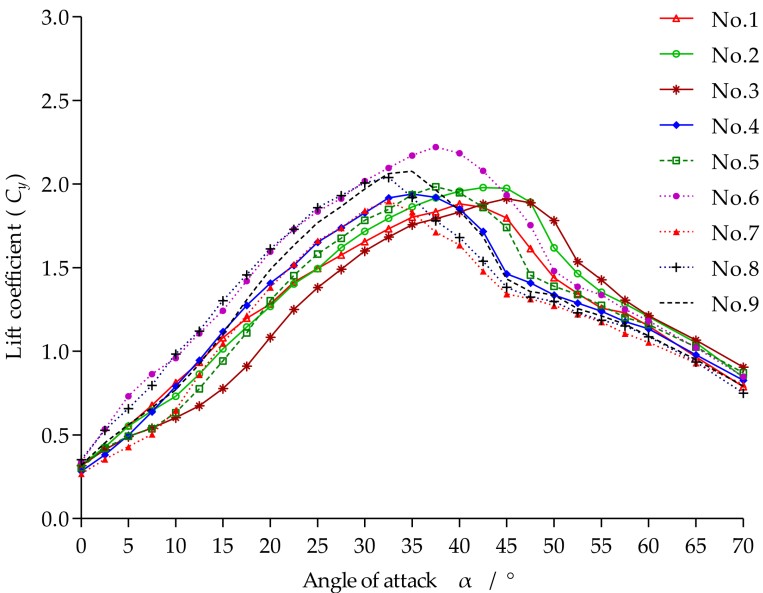

**Figure 8.** $C_y$ versus the angle of attack $\alpha$ for the nine otter board models.

Figure 9 presents curves of the lift–drag ratio $K$ versus the angle of attack $\alpha$ for the nine otter board models. The lift–drag ratio $K$ is highest for otter board model 8, being 4.052 ($\alpha$ = 17.5°), followed by otter board model 7, being 4.021 ($\alpha$ = 17.5°), and lowest for otter board model 3, being 2.917 ($\alpha$ = 22.5°). The angle of attack corresponding to the maximum lift–drag ratio of the nine models ranges from 15° to 22.5°, which is more than 10° smaller than the critical angle of attack.

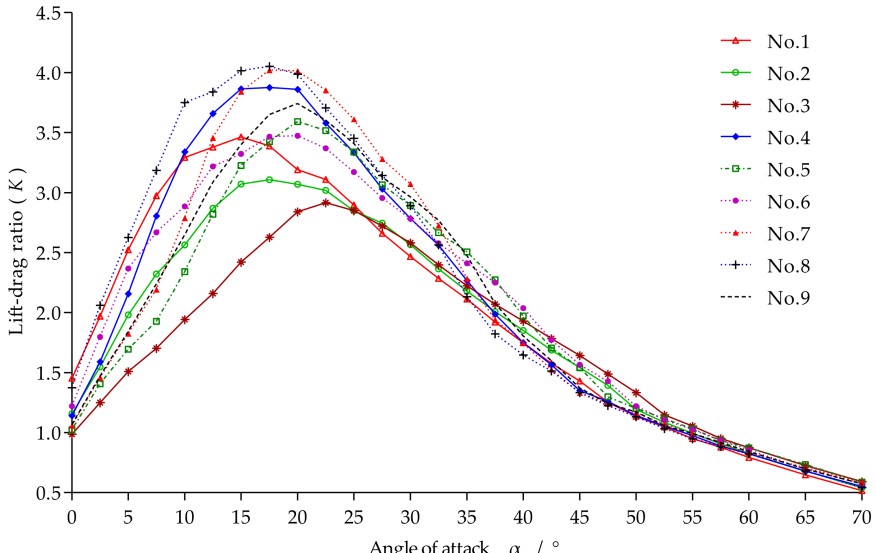

**Figure 9.** $K$ versus the angle of attack $\alpha$ for the nine otter board models.

### 3.2. Orthogonal Analysis

To better combine the three factors and their respective levels, the lift–drag ratio, a commonly measured parameter of the performance of an otter board, was used as an inspection index in orthogonal experimental analysis. According to the application design theory of otter boards, the selected working attack angle of the otter board is usually less than the critical attack angle of 2°–4° [21]. Referring to the range of the critical attack angle of the nine otter board models obtained in the previous analysis, the lift–drag ratio for a working angle of attack of 30° is used as an inspection index to calculate the scores at each level in this study (Table 6).

**Table 6.** Analysis of orthogonal testing at an angle of attack of 30°.

| No. | Level | Aspect Ratio | Level | Camber | Level | Gap Ratio | $C_x$ | $C_y$ | $K$ |
|---|---|---|---|---|---|---|---|---|---|
| 1 | I | 1.0 | I | 0.12 | I | 0.25 | 0.672 | 1.656 | 2.466 |
| 2 | I | 1.0 | II | 0.14 | II | 0.30 | 0.670 | 1.717 | 2.565 |
| 3 | I | 1.0 | III | 0.16 | III | 0.35 | 0.620 | 1.600 | 2.580 |
| 4 | II | 1.5 | I | 0.12 | II | 0.30 | 0.657 | 1.828 | 2.784 |
| 5 | II | 1.5 | II | 0.14 | III | 0.35 | 0.617 | 1.783 | 2.892 |
| 6 | II | 1.5 | III | 0.16 | I | 0.25 | 0.725 | 2.018 | 2.783 |
| 7 | III | 2.0 | I | 0.12 | III | 0.35 | 0.600 | 1.843 | 3.072 |
| 8 | III | 2.0 | II | 0.14 | I | 0.25 | 0.694 | 2.007 | 2.892 |
| 9 | III | 2.0 | III | 0.16 | II | 0.30 | 0.665 | 1.969 | 2.963 |
| | $I_j$ | 7.612 | | 8.323 | | 8.141 | | | |
| | $II_j$ | 8.459 | | 8.349 | | 8.313 | | | |
| | $III_j$ | 8.927 | | 8.326 | | 8.544 | | | |
| | $R_j$ | 1.316 | | 0.026 | | 0.404 | | | |

Note: $I_j$, $II_j$, $III_j$: sum of $K$ values at the same level; $R_j$: range; $C_x$: drag coefficient; $C_y$: lift coefficient; $K$: lift–drag ratio.

An orthogonal analysis is conducted to analyze the value of each factor and level. That is to say, by comparing $I_j$, $II_j$, and $III_j$, the effects of the factors on the experimental results can be distinguished, and the levels can be optimally combined.

For factor A (the aspect ratio), $I_A = 7.612$, $II_A = 8.459$ and $III_A = 8.927$, $III_A > II_A > I_A$, and it is seen that the optimal aspect ratio is 2.0. For factor B (the camber), $I_B = 8.323$, $II_B = 8.349$, and $III_B = 8.326$, $II_B > III_B > I_B$, and it is seen that the optimal camber of the fore wing is 0.14. For factor C (the gap ratio), $I_C = 8.141$, $II_C = 8.313$, and $III_C = 8.544$, $III_C > II_C > I_C$, and it is seen that the optimal gap ratio is 0.35. Without considering the interaction of the three factors, the optimal combination of levels is $A_3B_2C_3$.

The range is the difference between the maximum and minimum of the sum of each level for the same factor, which reflects the primary and secondary orders of each factor. Table 5 shows that $R_A > R_C > R_B$; i.e., the aspect ratio (factor A) is the most important factor, followed by the gap ratio (factor C) and finally the camber (factor B).

The above analysis reveals that the optimization result of the orthogonal experiment of the nine otter board models designed in this paper is $A_3B_2C_3$, with the aspect ratio strongly affecting the hydrodynamic performance of the otter board.

### 3.3. Numerical Simulation Verification

According to the orthogonal optimization results, the design and modeling of otter board model 10 are carried out (Table 7), and otter board model 7 has a higher lift–drag ratio (3.072) at an angle of attack of 30° is selected for comparison.

**Table 7.** Dimensions and structural parameters of the optimized model.

| No. | $L$/m | $b$/m | Aspect Ratio | Camber | Gap Ratio |
|---|---|---|---|---|---|
| 10 | 0.30 | 0.60 | 2.0 | 0.14 | 0.35 |

Note: $L$: chord; $b$: span.

Referring to the experimental conditions of the wind tunnel, the hydrodynamic performance of otter board models 7 and 10 was numerically simulated. The obtained coefficients are compared with experimental data for otter board model 7 (Figure 10); the average deviations of $C_x$, $C_y$, and $K$ between the numerical simulation and experimental results of otter board model 7 were approximately 8%, 5%, and 4%, respectively. The trends of the $C_x$–$\alpha$ curve, $C_y$–$\alpha$ curve, and $K$–$\alpha$ curve of otter board model 7 obtained in the numerical simulation are basically consistent with the results of the wind tunnel test. Figure 10 shows that the numerical $C_y$ of model 7 is lower than the test $C_y$ for most of the angle of attack range, and conversely, the simulated $K$ value is higher than the test data for most of the angle of attack range.

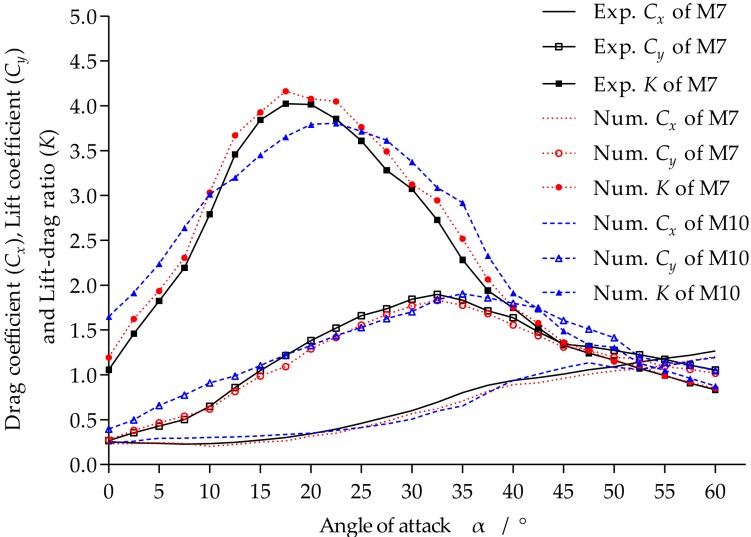

**Figure 10.** Comparison of the hydrodynamic performances of the otter board models.

The $K$–$\alpha$ curves of otter board models 7 and 10 in Figure 10 show that the simulated maximum $K$ value (4.239, $\alpha = 20°$) of otter board model 7 is higher than that of board model 10 (3.805, $\alpha = 22.5°$). After reaching a peak, the $K$–$\alpha$ of otter board model 7 decreases rapidly, whereas the $K$–$\alpha$ curve of otter board model 10 decreases gently. As the angle of attack increases to 30°, the $K$ value of otter board model 10 is higher than that of otter board model 7.

The numerical $C_y$–$\alpha$ curves of otter board models 7 and 10 in Figure 9 show that the maximum lift coefficient of otter board model 7 is 1.830 ($\alpha = 32.5°$), whereas that of otter board model 10 is 1.902 ($\alpha = 35°$). Therefore, the working angle of attack can be considered as 25°–30° (otter board model 7) and 27.5°–32.5° (otter board model 10). In the respective working ranges of the angle of attack, the lift coefficient and lift–drag ratio of otter board model 10 are higher than those of otter board model 7, showing better optimization.

## 4. Discussion

### 4.1. Aspect Ratio

Relevant studies have shown that for the same structure of an otter board, an increase in the aspect ratio reduces the eddy current area on the back of the otter board and thereby reduces the resistance and improving the lift–drag ratio [22,23]. The drag coefficient $C_x$, lift coefficient $C_y$ and lift–drag ratio $K$ of the nine otter board models at an angle of attack of 30° are plotted in Figure 11. Comparatively, an otter board with a larger aspect ratio (2.0) has a higher lift–drag ratio. Figure 9 shows that the three models with an aspect ratio of 2.0 have the top three maximum lift coefficients. The results of the orthogonal experiment reported in this paper show that the highest of the three levels of the aspect ratio factor is optimal.

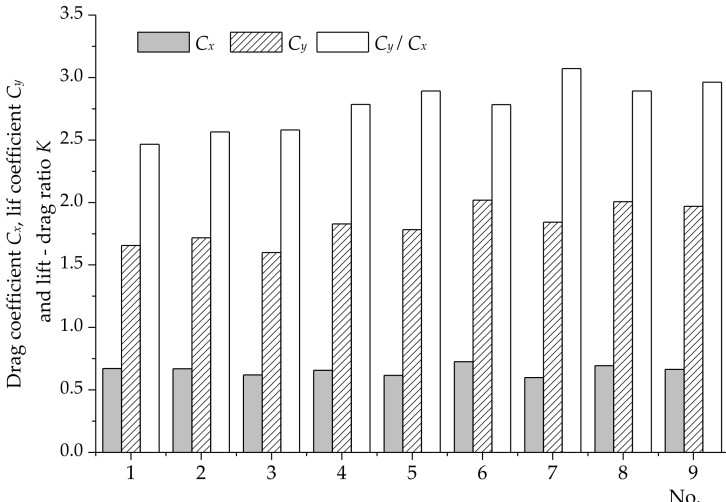

**Figure 11.** $C_x$, $C_y$, and $K$ of the otter board models for an angle of attack of 30°.

### 4.2. Camber

Disregarding the camber, otter board model 10 optimized in the orthogonal experiment has the same aspect ratio and gap ratio as otter board model 7. The camber of otter board model 10 is adjusted from 0.12 to 0.14 to that of otter board model 7. Figure 9 shows that an appropriate increase in the camber improves the lift–drag ratio of the otter board, and it also improves the lift within a certain range of the angle of attack. Figure 12 shows the flow state of otter board models 7 and 10 simulated at an angle of attack of 30°. It is seen that the eddy current area generated by otter board model 7 is larger, and the eddy current at the back end of the rear wing has been separated, while there is only a wingtip vortex on the front plate of otter board model 10, and the vortex has not yet separated. The separation of the vortices is responsible for the reduction in lift, while the larger area of the back vortex means that the drag will be higher, slowing the separation of the vortex at the back of the otter board, which usually improves the performance of the otter board [15]. Consistently, the flow diagram analysis in Figure 12 shows a better hydrodynamic performance of optimized otter board 10 with larger camber. This theory of optimizing the performance of the otter board by improving the wingtip vortex of the otter board has previously been confirmed [24].

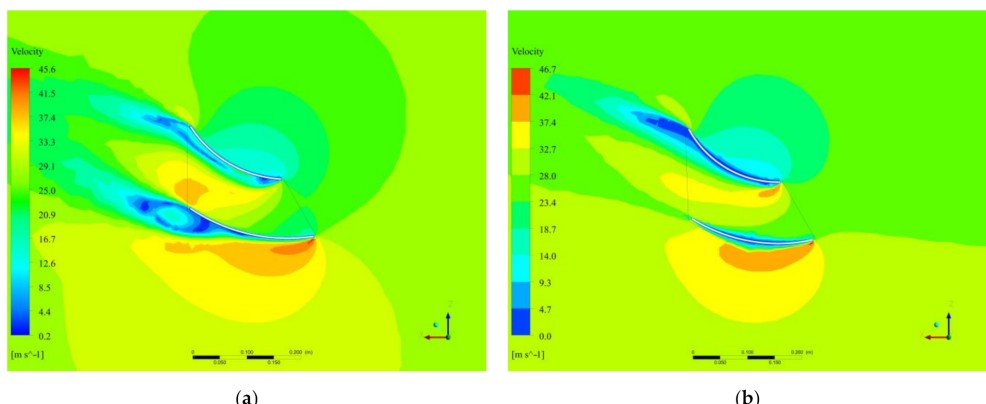

**Figure 12.** Flow state of otter board models 7 and 10 for an angle of attack of 30°: (**a**) Flow state of model 7; (**b**) Flow state of model 10.

### 4.3. Gap Ratio

The gap ratio of an otter board is usually measured using the horizontal or longitudinal gap between double-layer or multi-layer wings. In the present orthogonal experimental analysis, the effect of the gap ratio on the hydrodynamic performance of the otter board is greater than that of the camber. In Figure 11, otter board models 1, 6, and 8 with the

smallest gap ratio (0.25) are the three best performing models in terms of the drag coefficient, whereas otter board models 3, 5, and 7 with a larger gap ratio (0.35) perform worst. The analysis shows that the drag coefficient decreases with the gap ratio increasing. Equation (4) reveals that the drag coefficient is inversely proportional to the lift–drag ratio and the reduction of the drag coefficient can improve the lift–drag ratio, which explains why the highest level of the gap ratio factor is optimal in the orthogonal experiment conducted in this study.

## 5. Conclusions

An orthogonal experiment was conducted to optimize the structural parameters of double-vane otter boards and obtain better hydrodynamic performance. Analysis revealed the optimal combination of the aspect ratio (2.0), fore wing camber (0.16), and gap ratio (0.35), and that the importance of the factors decreased in the order of the aspect ratio, gap ratio, and camber. The analysis of experimental data showed that in terms of the three factors investigated, the otter board model with a larger aspect ratio had a higher lift-to-drag ratio, and an increase in the gap ratio reduced the resistance of the otter board model and thus affected the lift–drag ratio. The optimal levels of the aspect ratio and gap ratio were both the highest considered. Investigation of the wing camber showed that an increase in camber improved the lift coefficient and lift–drag ratio within a certain range. However, it also increased the drag coefficient, which reduced the lift–drag ratio, and the optimal camber factor was thus the second level considered. For the three factors of the orthogonal experiment in this paper, the aspect ratio and gap ratio were taken as the highest levels, and only the camber factor was selected at the second level. Future research can extend the levels of the aspect ratio and gap ratio for further optimization to explore double-vane otter boards with good hydrodynamic performance. Regarding the camber, a double-vane otter board with an airfoil structure may have better hydrodynamic performance. The experimental results provide a reference for the optimal design of the double-vane otter board.

**Author Contributions:** Conceived and designed the experiments, L.W. and X.Z.; performed the experiments, L.W. and G.Q.; analyzed and simulation, L.W. and Q.X.; writing—review and editing, L.W.; supervision, R.W.; funding acquisition, X.Z. All authors have read and agreed to the published version of the manuscript.

**Funding:** This research was funded by the National Key R & D Program of China (No. 2020YFD0901203).

**Institutional Review Board Statement:** Not applicable.

**Informed Consent Statement:** Not applicable.

**Data Availability Statement:** Not applicable.

**Acknowledgments:** We thank the staff at ECSF who assisted with the project, including Yongli Liu, Yu Zhang, Yongjin Wang, Aizhong Zhou, and Zhongqiu Wang, engineers, for their assistance in wind tunnel testing at the School of Nanjing University of Aeronautics and Astronautics.

**Conflicts of Interest:** The authors declare no conflict of interest.

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
