# Peer review of "Optimization of the Hydrodynamic Performance of a Double-Vane Otter Board Based on Orthogonal Experiments"

_jmse, doi:10.3390/jmse10091177_

Round 1
Reviewer 1 Report
In this work orthogonal experiment was conducted to study the effects of the aspect ratio, camber of the fore wing, and gap ratio of front and rear wing panels on the hydrodynamic performance of double-vane otter board. The optimal results of the orthogonal experiment were analyzed and verified in numerical simulation adopting the numerical computational fluid dynamics code ANSYS CFX.
The paper is worthy to be published in Journal of Marine Science and Engineering with some minor corrections/comments given below:
Mathematical model describing the fluid flow considered in the study should be added and explained in manuscript to help increase of understanding of readers about this work.
Details about model used are not provided. Why? As most of the readers may not be familiar with this model. And how it is incorporated in governing mathematical model of fluid flow?
The results are presented for the aspect ratio (1.0, 1.5 and 2.0), camber (0.12, 0.14 and 0.16) and gap ratio (0.25, 0.30 and 0.3). Are these valid for other values?
It is better to include the percentage deviations between the results obtained through experiment and numerical
What minimal percentage deviation was set by authors as a threshold to judge whether the results are in good agreement with experimental values?
Some future scope of the study should be added in the conclusion section.
Reviewer 2 Report
The authors optimize the shape of a piece of fishing equipment known as the otter board. They performed wing-tunnel testing of several different designs, and supplemented them with CFD simulations. The study is well-designed and the results are conclusive.
There is only one point that I did not quite understand. The authors write about hydrodynamic performance. Does the device operate in water? If so, why is it tested in a wind tunnel in an airflow (fig. 3)? Does the study respect the dynamic similarity? Otherwise, in the case if the authors are interested in the performance in air and not in water, why do they write "hydrodynamic" and not "aerodynamic"? I suggest that the authors should revise the manuscript to add clarity.
Author Response
Point 1: The authors write about hydrodynamic performance. Does the device operate in water? If so, why is it tested in a wind tunnel in an airflow (fig. 3)? Does the study respect the dynamic similarity? Otherwise, in the case if the authors are interested in the performance in air and not in water, why do they write "hydrodynamic" and not "aerodynamic"? I suggest that the authors should revise the manuscript to add clarity.
Response 1:
The otter boards are used for trawling in the sea, so write” hydrodynamics”.
The water tank experiment is a good way to carry out the hydrodynamic test of the otter board, but in the early days when there was no water tank equipment, and the principle of the expansion force of the otter board was the same as the liff the wing, the researchers tried to use the wind tunnel which is used for aircraft wing experiments to carry out the otter board experiment. After the construction of the water tank experiment facility, the water tank was begun to used for researching the otter boards.
For the method of studying the performance of the otter board, researchers had also conducted some comparative analysis of the similarities and differences between the wind tunnel and the water tank, and found that their results were highly consistent. The wind tunnel test has the characteristics of convenience, rapidity and accurate results, and now it is also an important way to carry out the hydrodynamic performance test of the otter board.
Although the medium of the wind tunnel is air and the medium of the water tank is water, according to the principle of model experiment - "Reynolds similarity", by adjusting the wind speed and the size of the model, the model experiment of the otter board can be controlled in the "automatic model areas”, which are described in the manuscript.
Please review, I have referred to your suggestions that the research description and literature about the wind tunnel experiment of the otter board are supplemented in "1 Introduction" of the manuscript.
Thanks a lot for your advice and guidance.

Reviewer 3 Report
No more further comments.
Author Response
English language and style are revised.
Reviewer 4 Report
The present work treats about an optimization of the hydrodynamic performance of a double vane otter board based on orthogonal experiments. The manuscript is interesting and well written. I recommend to publish it subjected to the following modifications:
- Improve the introduction section. More references must be included and a critical analysis about them. If possible, references of the Journal of Marine Science and Engineering would be beneficial for the journal.
- Have you performed a mesh size sensibility analysis? The Yplus is too high.
- Why have you chosen the e-epsilon turbulence model? Have you compared several turbulence models?
- Why have you set the turbulence intensity as 2.86 – 3.05? Justify. Idem regarding the viscosity ratio value of 5.
- Please, include a figure to illustrate the difference between experimental and numerical results in order to validate the numerical model.
